# The Resistance of Maize to *Ustilago maydis* Infection Is Correlated with the Degree of Methyl Esterification of Pectin in the Cell Wall

**DOI:** 10.3390/ijms241914737

**Published:** 2023-09-29

**Authors:** Yingni Huang, Yang Li, Kunkun Zou, Yang Wang, Yuting Ma, Dexuan Meng, Haishan Luo, Jianzhou Qu, Fengcheng Li, Yuanhu Xuan, Wanli Du

**Affiliations:** 1Specialty Corn Institute, College of Agronomy, Shenyang Agricultural University, Shenyang 110866, China; yingnihuang@163.com (Y.H.); ly20210129@163.com (Y.L.); z15524334236@163.com (K.Z.); wy21062107@163.com (Y.W.); 15140085901@163.com (Y.M.); dxmeng@syau.edu.cn (D.M.); luohs@syau.edu.cn (H.L.); qujz0220@syau.edu.cn (J.Q.); 2Rice Research Institute, Shenyang Agricultural University, Shenyang 110866, China; fengchengli@syau.edu.cn; 3College of Plant Protection, Shenyang Agricultural University, Shenyang 110866, China

**Keywords:** maize, common smut, cell wall, pectin, methyl esterification

## Abstract

Common smut caused by *Ustilago maydis* is one of the dominant fungal diseases in plants. The resistance mechanism to *U. maydis* infection involving alterations in the cell wall is poorly studied. In this study, the resistant single segment substitution line (SSSL) R445 and its susceptible recurrent parent line Ye478 of maize were infected with *U. maydis*, and the changes in cell wall components and structure were studied at 0, 2, 4, 8, and 12 days postinfection. In R445 and Ye478, the contents of cellulose, hemicellulose, pectin, and lignin increased by varying degrees, and pectin methylesterase (PME) activity increased. The changes in hemicellulose and pectin in the cell wall after *U. maydis* infection were analyzed via immunolabeling using monoclonal antibodies against hemicellulsic xylans and high/low-methylated pectin. *U. maydis* infection altered methyl esterification of pectin, and the degree of methyl esterification was correlated with the resistance of maize to *U. maydis*. Furthermore, the relationship between methyl esterification of pectin and host resistance was validated using 15 maize inbred lines with different resistance levels. The results revealed that cell wall components, particularly pectin, were important factors affecting the colonization and propagation of *U. maydis* in maize, and methyl esterification of pectin played a role in the resistance of maize to *U. maydis* infection.

## 1. Introduction

Common smut in maize is a global disease caused by *Ustilago maydis* (DC) Corda. High temperature and low moisture conditions are favorable for its development. *U. maydis* is a parasitic, biotrophic fungus and a plant pathogen, mainly infecting gramineous plants. Its infection can significantly decrease the yield of corn, wheat, barley, sorghum, and other staple crops [1]. One of the important noticeable features of *U. maydis* infection is enlarged tumors in the ears, and severe infection can lead to loss of the ear and a decrease in yield [2]. *U. maydis* does not directly cause plant cell death after infection; it gradually establishes a close symbiotic relationship with the infected host, which allows the host to remain alive during the whole life cycle of *U. maydis*. Smut in maize neoplasm can infect all young shoot tissues and induce tumor production, which in turn affects growth and developmental processes such as photosynthesis and sugar transport [3,4]. Under pathogenic infection, the internal receptor proteins in plants activate a series of defense response mechanisms upon sensing pathogenic stimuli to resist or remove the pathogens, and in turn attenuate their effects on plant growth and development.

The plant cell wall is the first line of defense against biotic factors. It is the recognition site and prevents the invasion of a majority of pathogens. In this way, it can activate immune responses in plants through cell wall proteins and induce the release of defense-related substances, thus inhibiting the spread of pathogens and playing an important role in resisting pathogenic infection [5]. The plant cell wall is composed of various complex polysaccharides including cellulose, hemicellulose, pectin, lignin, and abundant cell wall glycoproteins. Plant cell wall composition and structure vary not only among different species but also in different parts of the same organ [6,7]. When any pathogen invades, the cytoplasmic membrane in plants recognizes the pathogen-associated molecular patterns (PAMPs) on the surface of the pathogen and triggers defense responses. Damage-associated molecular patterns (DAMPs), such as the plant cell wall and cuticle fragments, are released to protect the host from pathogens [2]. Initially, it was thought to be an inert structure that surrounds all plant cells and provides support and protection. However, more recent studies have reported that it is a highly dynamic and plastic structure [8]. Studies have reported a specific mechanism in the cell wall that detects introduced modifier genes and regulates their expression to maintain a balance between the ratios of its components [9]. One study suggests that a key element of plasticity is involved in the mechanism of the maintenance of cell wall integrity (CWI); it detects the effects of genetic modifications and initiates the corresponding regulatory functions. More importantly, the CWI maintenance mechanism constantly monitors the functional integrity of the cell wall and readily initiates compensatory functions of the cell wall and cell metabolism. Dynamically, it regulates cell wall damage induced by biotic and abiotic stresses during cell morphogenesis, thereby maintaining CWI [10]. In summary, cell wall plasticity is crucial for plants to resist pathogenic invasion and limit the reproduction of pathogens in the host.

The plant cell wall is resistant to invasion by pathogens. Its ability to sense signals from pathogens, while acting as a signal transduction site, further activates the immune responses in plants [11]. In plant–pathogen interactions, the chemical composition or physical structure of the cell wall is changed, which is known as cell wall resistance [12]. At the site of pathogenic infection, structural changes in callose, lignin, cellulose, and pectin can improve or strengthen cell wall resistance in plants. After invading plant cells, pathogenic fungi produce cytoplasmic agglutination below the aphidaspora inducing the halo reaction and forming a hard and solid deposit on the nearby cell wall, which is called the mastoid structure [13]. Moreover, pectin, callose, and lignin play important defensive roles in the interactions of the cell wall with pathogens. Typically, pectin, as one of the main sources of DAMPs, is a complex polysaccharides homogalacturonan in the primary cell wall that contains linear molecules with approximately 10–16 α-1,4-D-galacturonic acid (GalpA) residues. Homogalacturonan (HG) consists of linear polymers of α-1-4-linked residues with methyl esterified (C6) and acetylated (C2/C3) forms that regulate cell wall plasticity while controlling plant cell growth and defense [14,15]. During an invasion of pathogens, HG can be degraded into oligogalacturonides (OAGs). Studies have reported that OAGs are released as a DAMP signal, which is recognized by wall-associated kinase1 (WAK1) thereby activating immune responses in plants [16,17]. In addition, OAGs trigger the inhibition of hormone-induced genes [18]; therefore, they play a role in balancing stress response and development. Shorter OAGs induce most immune responses but do not produce reactive oxygen species, suggesting that OAGs with different degrees of aggregation may trigger different responses [19]. However, given the complexity and availability of pectin polysaccharides, other pectin-derived carbohydrate-based molecules may also contribute to the activation of cell wall-degrading enzymes (CWDEs) [20]. It is evident that pectin plays an important regulatory role in plant defense against diseases.

In this study, we aimed to assess the defense mechanism in maize cells under *U. maydis* infection via alterations in the cell wall. The resistant and susceptible inbred lines of maize, namely, R445 and Ye478, respectively, were used to identify the complex physiological and biochemical changes after *U. maydis* infection. The main aims of this study were (1) to investigate the changes in the cell wall components of resistant and susceptible lines after *U. maydis* infection, (2) to visualize the changes in the cell wall using immune antibody markers, and (3) to validate the function of high/low-methylated pectin in 15 maize inbred lines with different resistant levels to *U. maydis*. This study will enhance the understanding of the molecular mechanism of interaction between maize and *U. maydis*.

## 2. Results

### 2.1. Identification of Disease-Resistant Phenotype in the Resistant and Susceptible Maize Plants Infected with U. maydis

The phenotypes of susceptible and resistant maize lines, Ye478 and R445, respectively, were regularly monitored after infection with *U. maydis* SG200 from 0 to 12 days postinfection (dpi). From 0 to 2 dpi, the appearance of the leaves was not significantly different between Ye478 and R445. At 4 dpi, the transparency of the leaves was changed, and chlorotic lesions were observed at the infection site in Ye478. However, no such typical symptom was observed in R445 at 4 dpi. At 8 dpi, more severe, massively aggregated tumors appeared in R445, and the number and size of tumors were significantly reduced compared with Ye478 at 8 dpi. At 12 dpi, in Ye478, several contiguous sheet intercalation tumors appeared, whereas R445 exhibited fewer and smaller tumors (Figure 1A). Three independent biological replicates including 300 infected seedlings per line were assessed for the symptoms of *U. maydis* infection. Disease index data revealed that Ye478 exhibited chlorosis; no symptoms occurred in only 16.7% of samples, and aggressive tumors occurred in 48.7% of samples. R445 exhibited chlorosis; no symptoms occurred in 47.2% of samples; moreover, no aggressive tumors were present in any samples (Figure 1B). Laser scanning confocal microtechnic technology was used to investigate the response of the leaves to infection. The fungal hyphae growth in the infected leaves was assessed and visualized using WGA-AF488/propidium iodide co-staining at 0, 2, 4, 8, and 12 dpi. The phenotypes were similar in Ye478 and R445 up to 2 dpi. However, the hyphae exhibited proliferation at 4 dpi in Ye478 but at 8 dpi in R445. Compared with R445, Ye478 exhibited greater number of and larger tumors in the leaves at 12 dpi (Figure 1C). To further study the disease progression in Ye478 and R445, relative fungal biomass was assessed by quantifying fungal DNA using qPCR. At 2 and 4 dpi, no significant difference was observed between the two lines. However, at 4 dpi, relative fungal biomass was approximately twice in Ye478 what is was in R445. Moreover, from 8 to 12 dpi, the relative fungal biomass of *U. maydis* significantly (*p* ≤ 0.01) increased in Ye478 but only slightly increased in R445, along with the accumulation of mature teliospores (Figure 1D).

The changes in cell morphology induced by *U. maydis* were observed using paraffin sectioning at 4 to 12 dpi. The size of the upper and lower epidermis was generally bigger in Ye478-CK than in R445-CK (Figure 2). At 4 dpi, no significant change in the leaf cells was observed in both lines; both upper and lower epidermis cells were distributed evenly, and bundle sheath cells were normal in number. A blue-green lignified layer was observed in the epidermis of R445-Inf but not in that of Ye478-Inf (Figure 2G,J). At 8 dpi, in Ye478-Inf, the epidermis cell wall was thickened; the number of bundle sheath cells was increased; cell swelling was observed; and surface area expanded after *U. maydis* infection. However, the bundle sheath cells were intensively divided in R445-Inf (Figure 2H,K). At 12 dpi, in Ye478-Inf, bundle sheath cells continuously divided and the mesophyll cell number was approximately doubled; bundle sheath cells continuously expanded, and their structural features were blurred during tumor maturation. However, the stress responses in R445-Inf were less severe than those in Ye478-Inf (Figure 2I,L).

### 2.2. Analysis of Cell Wall Components and CWDEs after U. maydis Infection

The main components of the cell wall are cellulose, hemicellulose, pectin, and lignin. Their contents were compared between the treatment and control groups of Ye478 and R445. The contents of cellulose, hemicellulose, pectin, and lignin increased in the treatment and control groups of Ye478 and R445 as the dpi increased; particularly, lignin content remarkably increased after 4 dpi in both groups (Figure 3). Cellulose content increased by 7%–10% and hemicelluloses content increased by 10%–15% at 4 and 8 dpi after *U. maydis* infection compared with the control (Figure 3A,B). Pectin content remained unchanged at 0–2 dpi but significantly increased at 4 dpi in the treatment group of both Ye478 and R445, but only in R445 at 12 dpi, suggesting that pectin played a role during the infection process (Figure 3C). In addition, lignin content increased by approximately two-fold from 4 to 12 dpi in both Ye478 and R445. This indicated that lignin production was promoted to resist pathogen attack (Figure 3D). In Ye478-Inf and R445-Inf, pectin methylesterase (PME) activity steadily increased from 2 to 8 dpi. From 8 to 12 dpi, it decreased slightly in the treatment groups compared with the control groups. This indicated that pectin may play a role in *U. maydis* infection (Figure 3E).

Further, the changes in the contents of cell wall components and the activities of their degrading enzymes during disease development were studied. The activities of CWDEs including α-L-arabinofuranosidase, β-galactosidase, β-glucosidase, and β-1, 4-glucosidase were measured. The activities of α-L-arabinofuranosidase and β-galactosidase in Ye478-Inf and R445-Inf gradually increased as the dpi increased and were the highest at 8 and 12 dpi, respectively (Appendix A). Similarly, the activities of β-glucosidase and β-1, 4-glucosidase in Ye478-Inf and R445-Inf exhibited a similar trend of first increasing and then significantly decreasing and were the highest at 4 dpi (*p* < 0.05) during the course of disease. The results indicated that CWDEs play a role in helping hyphae penetrate the cell wall of maize.

### 2.3. Assessment of Changes in Cell Wall Components and Structure via Immunofluorescence Analysis

The changes in the cell wall components and structure were investigated using a series of monoclonal antibodies, namely, LM10 (xylan), JIM5 (low-methylated pectin), and LM20 (high-methylated pectin), which revealed the subcellular distribution of their antigenic epitope; in general, control groups exhibited more intensive signals than treatment groups. Hemicellulose appeared mainly in bundle sheath cells; weaker and stronger signals were observed in the epidermis of Ye478-Inf and R445-Inf at 4 dpi, respectively. Immunolabeling in Ye478-Inf was slightly stronger in the bundle sheath cells but weaker in the epidermis compared with Ye478-CK at 12 dpi. However, R445-Inf exhibited more intensive signals in both bundle sheath cells and the epidermis at 8 dpi. Weaker immunolabeling with LM10 (for hemicellulose) was observed in Ye478-Inf than Ye478-CK, and it was relatively weaker in R445-Inf than R445-CK at 12 dpi (Figure 4A).

Immunolabeling with the JIM5 antibody revealed that low-methylated pectin mainly appeared in the mesophyll cells. A weak signal was observed in the epidermis in Ye478-Inf and R445-Inf at 4 dpi. Moreover, Ye478-Inf and R445-Inf exhibited weaker signals than Ye478-CK and R445-CK, respectively, from 4 to 12 dpi. The size of tumor increased as dpi increased. A weak labeling signal was observed scattered from 4 to 12 dpi; however, the signal in the mesophyll cells clearly enhanced at 8 dpi. A continuous weak signal was observed in both Ye478-Inf and R445-Inf and nearly disappeared at 12 dpi, particularly in R445-Inf. The results were consistent with the results of the PME activity (Figure 4B).

At 4 dpi, immunolabeling with LM20 antibodies indicated that the signals were relatively stronger in Ye478-Inf and R445-Inf, particularly in the bundle sheath cells, epidermis, and mesophyll cells. The signals were stronger in Ye478-CK than in R445-CK. However, at 8 dpi, a slightly weaker signal was observed in the bundle sheath cells, epidermis, and mesophyll cells in Ye478-CK than in R445-CK, respectively. At 12 dpi, a significantly weaker signal was observed in the bundle sheath cells, epidermis, and mesophyll cells in Ye478-Inf than in Ye478-CK. Consequently, the signal decreased significantly more in R445-Inf than in R445-CK, and a labeled cell wall profile was observed, which was consistent with the results of PME activity (Figure 4C).

### 2.4. Analysis of Phenotype and PME Activity in the Maize Seedlings with U. maydis Infection

To determine the role of the cell wall in the case of *U. maydis* infection, 15 maize inbred lines with different resistant levels [susceptible, partially susceptible, and resistant (5 inbred lines of each)] were inoculated with *U. maydis* at the seedling stage. This experiment included three independent biological replicates under controlled conditions. For each line, an average 100 plants were scored according to the symptoms. Interestingly, the susceptible maize lines (Dong6002, HuangC, KL4, P007, and 391) originated from five different heterotic groups; all five partially susceptible lines (Xin444, Dan598, 1028, Longxi69, and Cheng351) originated from at least three groups, whereas all five resistant lines (444, K12, Ji1037, Dan6263, and Sui601) originated from four different groups (Appendix A; Figure 5A). This indicated that these different inbred lines had different sources (heterotic groups) of resistance to *U. maydis*.

To further characterize the change in pectin in various maize lines after *U. maydis* infection, PME activities were measured. At 8 dpi, the PME activity in the leaves was significantly higher after *U. maydis* infection than in controls. Some susceptible (Dong6002, HuangC, KL4, and P007), partially susceptible (Dan598 and Cheng351), and resistant (444 and Ji1037) lines displayed significantly different demethylation of pectin during disease development, an LSD test showed that there was significant otherness between the susceptible and resistant groups at the 0.05 level (Figure 5B).

### 2.5. Changes in Pectin Methyl Esterification in Maize Lines during Host–Pathogen Interactions

Based on the resistance level, origin, good growth, and PME activity especially, two lines were selected from each group of inbred lines (susceptible, partially susceptible, and resistant), namely, Dong6002, P007, Dan598, Cheng351, 444, and Ji1037, for the qualitative analysis of pectin at 8 dpi using low/high-methylated pectin immunolabeling antibodies JIM5 and LM20. Immunolabeling with the JIM5 antibody was weaker in all lines after infection than in the control. Moreover, partially susceptible and resistant lines exhibited relatively stronger immunolabeling than susceptible lines after the infection compared with the control. Immunolabeling with the LM20 antibody exhibited stronger signals in the susceptible and partially susceptible lines but a weaker signal in the resistant lines after the infection. In contrast, partially susceptible and resistant lines exhibited weaker immunolabeling than susceptible lines before *U. maydis* inoculation (Figure 6). Therefore, it can be inferred that the degree of pectin methyl esterification significantly affected the ability of *U. maydis* to infect maize.

### 2.6. Changes in the Expression of PME-synthesis-related Genes in Maize Lines Infected with U. maydis

The PME-synthesis-related genes of maize, *Arabidopsis*, and rice were subjected to conjoint analysis using MegaX and Tbtools software. Seven genes with similar structures and functions were selected for qRT-PCR according to the comprehensive analysis and our previous transcriptome data (Appendix A; Appendix A). The seven genes were upregulated after *U. maydis* infection at 8 dpi in the 15 inbred lines with different resistance levels. The expression of *Zm00001d009899*, *Zm00001d009057*, *Zm00001d040649*, and *Zm00001d028819* increased approximately 3–9 times after *U. maydis* infection. In general, in the expression level of *Zm00001d009057*, *Zm00001d040649*, *Zm00001d051503*, and *Zm00001d030712* exist inter-group differences between the susceptible, partially susceptible, and resistant lines (Figure 7B–E). The expression of *Zm00001d051503*, *Zm00001d030712*, and *Zm00001d010793* was relatively higher by approximately 0.4–2 times after the infection than the control (Figure 7D–F). These results suggested that pectin possibly contributed to the resistance of maize to *U. maydis*.

## 3. Discussion

### 3.1. U. maydis Induced Changes in Maize Leaf Cell Structure

Common smut caused by *U. maydis* can hinder the growth of plants and reduce yield, causing serious economic losses. The cellular structure of plants is affected due to pathogenic infection, and cell composition is altered. The dynamic process of sugarcane smut development was observed using paraffin sectioning, which revealed that the resistant lines prevented further fungal invasion by enhancing cell wall thickness [21]. Previous studies have reported that the vascular sheath cells began to divide after 4 dpi, and the mesophyll cells clearly exhibited edema after 6 dpi, along with tumor formation. After 8 dpi, the epidermal cells gradually became larger, and the number of vascular sheath cells decreased after *U. maydis* infection from 36 h to 13 days [22]. Similarly, Lin [23] reported the phenomenon of vascular sheath cell division and mesophyll cell enlargement by infecting maize leaves with FB1×FB2 strains. However, when these leaves were infected with the FB1Δnlt1×FB2Δnlt1 mutant strain, cell division and mesophyll cell enlargement were not observed. Zhang [24] analyzed the differences in the tissues of resistant lines and observed the division and proliferation of vascular sheath cells. Our results indicated that a large number of new cells were generated around the vascular bundle sheath of the leaves of the susceptible lines, resulting in the swelling of the leaf structure (Figure 2H,I). R445 also produced new cells but in smaller numbers (Figure 2K,L). A similar phenomenon was reported by a previous study; the cells around the sheath of the vascular bundle in the leaves were greatly swollen, and the tumors formed were mainly due to swollen mesophylls in the resistant lines after *U. maydis* infection [22]. In Y478, the thickness of the epidermal cell wall significantly increased after tumor formation; the number of epidermal cells did not change significantly; the number of vascular bundle sheath and primary vein cells significantly increased, and the area of the vascular bundle sheath increased (Figure 2H,I). However, in R445, no significant increase in cell wall thickness; no significant change in the number of vascular bundle sheath, primary vein, and epidermal cells; and a significant increase in vascular bundle sheath area were observed (Figure 2K,L). In conclusion, the thickness of the epidermal cell wall might play an important role in the resistance to *U. maydis*.

### 3.2. The Changes in Cell Wall Components after U. maydis Infection

The cell wall around plant cells forms a natural barrier against various stresses and pathogens and determines cell expansion and shape [10]. A series of changes are induced in the cell wall composition and structure to prevent pathogenic invasion. Pathogenic invasion alters cell wall components or modifies the physical structure to confer resistance, which is called cell wall resistance [11]. For pathogens, destroying plant cell walls is a significant challenge. It is well known that the cell wall is mainly composed of cellulose, hemicellulose, pectin, and lignin [25]. Among these components, cellulose is generally composed of microfibril and is the most abundant polysaccharide polymer in plants that affects plant growth and defense responses. Plants with impaired cellulose synthesis exhibit severe developmental defects, i.e., stunting and significantly decreased yield. In most cases, cellulose-synthesizing mutants exhibited improved resistance; for example, in *A. thaliana*, mutations in three types of cellulose synthase (CESA) subunits CESA 4, CESA 7, and CE-SA 8 enhanced resistance to *Ralstonia solanacearum* and *Plectosphaerella cucumerina* [26]. However, in a few rare cases, impaired cellulose synthesis leads to reduced plant disease resistance. The gene encoding cellulose-synthase-like D2 in barley (*HvCslD2*) was silenced through RNA interference (RNAi), which resulted in decreased cellulose content in the epidermal cell wall; its mastoid structure was more easily penetrated by powdery mildew pathogens [27]. In our study, a significant amount of cellulose was accumulated in the tumors infected with *U. maydis* (Figure 3A). A possible reason for the increase in cellulose content was that numerous proliferating cells produced more cellulose in the tumors. Evidence suggested that the alterations in xylan or xylglucan, which are the main components of hemicellulose, directly affect the resistance of *Arabidopsis* to pathogens [28]. The de-etiolated3 (det3), pom-pom1 (pom1), and ectopic lignification1 (eli1) mutants exhibited abnormalities in lignin biosynthesis and regulation [29]. The secondary cell wall-specific cellulose synthase genes, namely, irregular xylem1 (IRX1) and IRX3, significantly decreased cellulose content [30]. Gene-encoding α-xylosidase (*Atxyl1*) was silenced, after which, remarkable changes in xyloglucan composition were observed, along with alterations in the growth pattern [31]. In this study, we observed that the resistant inbred line R445 contained more hemicellulose components than Ye478. As the *U. maydis* infection progressed, more proliferative cells were formed, eventually promoting hemicellulose accumulation (Figure 3B). When a pathogen penetrates the outermost cuticles of plant, pectin is the first barrier to prevent the invasion. Pectin is highly conserved and forms hydrated networks interacting with other cell wall components. This determines the pores in the cell wall; provides a charged surface that regulates cell wall pH and ion balance; regulates intercellular junctions; and transfers signals or recognition molecules from symbiotic organisms, pathogens, and insects to plant cells [10,32]. Pectin has four polysaccharide domains, namely, homogalacturonan (HG), rhamnogalacturonan I (RGI), rhamnogalacturonan II (RGII), and xylogalacturonan (XGA) [33]. In *Arabidopsis*, the *gae1* and *gae6* double-mutant plant exhibited an impaired biosynthesis of HG and RG-I, which clearly improved their resistance to *Pseudomonas syringae* and grey mold [34]. Additionally, *pmr5* and *pmr6* mutants of *Arabidopsis* exhibited increased pectin content, which enhanced the resistance of *Arabidopsis* mutants by limiting the penetration of powdery mildew fungi [35,36]. It was inferred that pectin participates in regulating plant disease resistance by changing its content. In this study, a similar trend was observed, i.e., pectin content significantly increased after 4 dpi in both Ye478-Inf and R445-Inf (Figure 3C). As a phenolic polymer, lignin content is generally positively correlated with plant disease resistance. When a plant is infected with pathogens, immune systems are immediately activated; the lignification degree and mechanical strength clearly improve, which result in the significantly enhanced tolerance of the cell wall to CWDEs released by pathogens [13,26]. Rice transcription factor *OsMYB30* promoted the expression of *Os4CL3* and *Os4CL5* genes related to lignin biosynthesis and caused the accumulation of lignin subunits G and S, during which sclerenchyma cells were strengthened to withstand the invasion of *Magnaporthe grise* [37]. Similarly, the results of this study revealed that the lignin-rich inbred line exhibited considerably high and stable horizontal resistance to pathogens. In other words, lignin content and plant disease resistance were significantly positively correlated (Figure 3D). However, a negative correlation also exists between lignin content and disease resistance; for example, the default expression of a gene encoding hydroxycinnamoyl-CoA—shikimate hydroxycinnamoyl transferase (HCT)—could decrease lignin content to enhance the resistance of *Medicago sativa* L. to anthracnose [38]. Therefore, lignin is known to play important roles in plant defense responses against pathogens and in plant–pathogen interactions.

### 3.3. The Function of PME in the Plant–Pathogen Interaction

The activity of PME is regulated by pectin methylesterase inhibitors (PMEIs). It is reported that PMEIs have originated from PME for the spatio-temporal coordination of the expression model to regulate methyl esterification degree of HG [39,40]. Moreover, the methyl esterification status of pectin positively correlates with plant–pathogen interactions. Pectin de-esterification directly influences the function of the cell wall barrier and bacterial CWDEs and further influences the growth of hyphae and the colonization of pathogens [41]. The pectin esterification state is regulated by PME and PMEI, and pectin with various degrees of methylation is produced [9,10]. Oligogalacturonide (OGAs) produced by PME de-esterification could serve as the signal for DAMPs to activate the immune response in plants [42]. After *Rhizoctonia solani* Kuhn infection, genes related to pectin metabolism were analyzed using rice transcriptome. A candidate gene *AG1IA_04727* encoding polygalacturonase (PG) was significantly upregulated during the infection, which suppressed sheath blight disease [43]. Interestingly, the activity of PME increased (Figure 3E) identically that of other four CWDEs increased (Appendix A). In plants, PME can hydrolyze the methyl ester group in HG and release methanol and proton. This either forms Ca^2+^ bonds or the so-called “egg-box” model structure to form gel and improve plant resistance, provides a degradation target for other pectin-degrading enzymes to reduce the resistance by effecting the cell wall structure and hardness, or acts as an activated substance to be recognized by the downstream defense response [39,44]. Furthermore, the expressions of PME and PMEI are strictly controlled by salicylic acid (SA), ethylene (Eth), and jasmonic acid (JA) [45]. Bano [46] analyzed cis-acting elements of the PMEI promoter of soybean and reported that SA cis-regulatory elements existed in GmPMEI, which provided the resistance to *Fusarium acuta* (*F. oxysporum*) and *Phytophthora soya* (*P. sojae*) with a defense mechanism via SA resistance pathway. Lionetti [47] identified that the expression of *AtPMEIs* is strictly regulated by JA and ET signaling during *Botrytis* infection in *Arabidopsis*, which upregulates the defense-related genes to limit PME activity, resulting in higher pectin methylation and abnormal functioning of the CWDEs secreted by pathogens [48]. In our previous study, at the early stage of *U. maydis* infection, the contents of jasmonic acid-isoleucine (JA-ILE), 1-aminocyclopropane-1-carboxylic acid (ACC), and SA were extremely significantly increased in Ye478, confirming that hormones are required for cross-linking with PMEIs to fine-tune the biosynthetic homeostasis of pectin polysaccharides [1]. Even if the expression of gene-encoding PME synthase increased (Figure 7), the activity of PME still increased under the cross-linking effect of PMEI and hormones.

### 3.4. Resistance to Pathogens Correlates with the Degree of Pectin Methyl Esterification 

To determine the role of pectin components in maize under *U. maydis* infection, the low-methylated pectin antibody JIM5 and the high-methylated pectin antibody LM20 were used for immunolabeling in Ye478 and R445. The result revealed that low-methylated pectin was mainly distributed in the vascular bundle sheath, primary leaf vein, mesophyll cells, and epidermis (Figure 4B). High-methylated pectin was mainly distributed in the bundle sheath and mesophyll cells (Figure 4C). In the later stage of tumor formation, the signal for low-methylated pectin in the susceptible line Ye478 was significantly stronger than that in the resistant line R445, whereas the signal for high-methylated pectin was stronger in the tumors of R445 (Figure 4B,C). The degree of pectin methyl esterification is reported to be relevant in the regulation of plant disease resistance. Raiola [49] reported that high-methylated pectin exhibits excellent endurance to CWDEs, thus conferring disease resistance to plants. Similarly, in our study, a higher pectin level was detected in R445-Inf, and Ye478-Inf and R445-Inf exhibited a higher signal for JIM5 (for low-methylated pectin) (Figure 4B) and LM20 (for high-methylated pectin), respectively (Figure 4C). These results are consistent with those of Lionetti [41] and Liu [50]. Importantly, the susceptible line exhibited stronger signals for low-methylated pectin after the formation of tumors under *U. maydis* infection. It was speculated that tumor formation may lead to an enhancement in the degree of lignification around the tumor, in which the tumor site is enriched in pectin and exhibits high levels of low- and high-methylated pectin.

The activity of PME gradually increased as the dpi extension; however, the increased PME activity could promote the pectin methyl esterification level. This demonstrated that the degree of pectin methyl esterification significantly changed during *U. maydis* infection; a similar observation was reported in a tobacco mosaic virus (TMV) infection in tobacco. Therefore, the reduced expression of PME slows the progression of viral infections [51]. Lionetti [47] reported that increased pectin esterification in *Arabidopsis* led to increased resistance to *B. cinerea* and *P. carotovorum* via overexpression of pectin methylesterase inhibitors AtPMEI-1 or AtPMEI-2. Actually, PME is the enzyme modulating the degree and pattern of methyl esterification of HG, which is delicately tuned to adjust the HG properties according to plant growth and development [52]. Generally, PME demethylates HG in a blockwise pattern, and PMEI modifies the distribution of methyl esters within the HG polymer by shifting the blockwise pattern to a random pattern, which was even worse when under a pathogen attack [53,54,55,56]. Wiethölter [57] reported that in the wheat–stem rust interaction, different patterns of HG methyl ester groups resulted in differential degradation of HG and different OG products and conferred different susceptible and resistant activities to near-isogenic wheat lines. These degraded small fragments, OG, or other cassette of carbohydrate-based molecules act as a danger signal to activate defense mechanisms and make the defense gene or protein function the same as HG [58]. Similarly, when the expressions of two glucuronate 4-epimerases (GAEs) family members *GAE1* and *GAE6* (which catalyze the formation of the key components of pectin from the precursor UDP-D-glucuronic acid) were repressed by a *P. syringae* pv *maculicola* infection in *A. thaliana*, the gae1 and gae6 double mutation significantly altered the OG signaling pathway and led to restricted soluble pectin release during pathogenic attack [34]. Overexpressed *CaPMEI1* in *Arabidopsis* could lead to a higher methyl esterification level that enhanced resistance to biotic and abiotic stresses [59]. These results demonstrated that the degree and pattern of pectin methyl esterification plays a crucial role in plant–pathogen interactions. The resistance conferred by pectin depended on the degree of esterification.

To further verify the relationship between the degree of pectin methyl esterification and disease resistance in maize, the stability of pectin methylase in 15 inbred lines with different resistant levels was measured at the seedling stage. The results revealed that PME activity in most susceptible lines significantly increased after *U. maydis* infection (Figure 5B). Two typical lines were selected from each of the 5 susceptible, partially susceptible, and resistant lines for immunolabeling with JIM5 and LM20 (high- and low-methylated pectin antibodies, respectively). The results revealed that the resistant lines exhibited relatively strong signal for low-methylated pectin. The signal for low-methylated pectin was weakened in the partially susceptible and susceptible lines; however, the signal for high-methylated pectin was enhanced in the susceptible lines (Figure 6). At the same time, several genes related to pectin methylesterase synthesis were upregulated after *U. maydis* infection, thereby reducing the degree of pectin methyl esterification. This was also an important reason for the enhancement of the signal for low-methylated pectin (Figure 7). Among them, *Zm00001d009057*, *Zm00001d040649*, and *Zm00001d009899*, which may play a role in the early infection response of *U. maydis*, were identified from the transcriptome data in our previous study [1]. The pectin methyl esterification level plays a crucial role in enhancing plant disease resistance. Additionally, in near-isogenic lines of wheat, the lines susceptible to the stem rust fungus *Puccinia graminis* exhibited an accentuated blockwise pattern distribution of methyl esters compared with resistant lines [57]. Therefore, we hypothesized that the large number of tumors formed in the susceptible lines caused the cell wall to thicken, resulting in the decrease in low-methylated pectin content but the increase in high-methylated pectin content to stiffen the cell wall and loosen the connection of different components of cell wall. Le Cam [60] reported that the susceptibility of carrot cultivars with a similar pectin content to *Mycocentrospora acerina* was affected to various degrees as per the pattern of esterification. Hence, the degree of pectin methyl esterification was one of the important factors causing the varying degree of resistance to *U. maydis* in maize.

## 4. Materials and Methods

### 4.1. Plant Materials and U. maydis Inoculation

The susceptible inbred line Ye478 was used as the recipient genetic background. The resistant chromosome segment substitution line (SSSL) R445 was used for the evaluation of resistance phenotype and the determination of cell wall components and immune antibody markers. The R445 line contained only one segment of donor chromosome derived from Qi319 marked by umc1528 and umc1489 introgressed into Ye478 (Appendix A, Appendix A). In the control group, the leaves of Ye478 and R445 were inoculated with ddH_2_O (labeled as Ye478-CK and R445-CK). In the treatment group, the leaves were infected with *U. maydis* (labeled as Ye478-Inf and R445-Inf). We selected 15 maize inbred lines with different resistant levels, i.e., susceptible, partially susceptible, and resistant to assess the role of the cell wall in *U. maydis* infection (Appendix A). LSD (Least Significant Difference) tests were used to compare 15 maize inbred lines in three treatment groups. All plants were grown in a phytotron (light/dark 14 h at 28°C/10 h at 22°C, relative humidity 50%).

*U. maydis* strain SG200 was cultured overnight in liquid YEPSL (0.4% bactopeptone, 1% yeast extract, and 0.4% sucrose) on a rotary shaker (200 rpm) at 28°C, followed by centrifugation at 5,000 rpm for 5 min. The cell pellet was resuspended in ddH_2_O to obtain cell suspension with OD_600_ = 1 [61].

Three-week-old maize seedlings were inoculated with *U. maydis* as follows: 2 mL of the cell suspension was injected into the whorls at the six-leaf stage using a syringe as described previously [2]. Further, 100 plant leaves were collected at 0, 2, 4, 8, and 12 dpi with three independent biological replicates. All the plant materials and the *U. maydis* strain SG200 are preserved at the Specialty Corn Institute, College of Agronomy, Shenyang Agricultural University, China.

### 4.2. Analysis of Cell Wall Components, PME, and CWDEs

The leaves collected at 0, 2, 4, 8, and 12 dpi were inoculated with *U. maydis* or ddH_2_O. Cellulose, hemicelluloses, lignin, and PME were isolated using a Cellulose (CLL) Content Assay Kit (Catalog Number AKSU007C), Themicellulose Content Assay Kit (Catalog Number AKSU008C), and Lignin Content Assay Kit (Catalog Number AKSU010U) (Beijing BOXBIO Biotechnology Co., Ltd., Beijing, China), respectively, and plant pectin methylesterases (PME) ELISA Kit (Catalog Number AD9217, ADANTI Anti-biological Technology Co., Ltd., Wuhan, China) as per the manufacturer’s protocol and quantified using Thermo Scientific Microplate Reader (1510, Vantaa, Finland).

The specific CWDEs including α-L-arabinofuranosidase, β-galactosidase, β-glucosidase, and β-1,4-glucosidase were extracted from the leaf samples of Ye478 and R445 lines collected at 0, 2, 4, 8, and 12 dpi using an α-L-Arabinofuranosidase (α-L-Af) Activity Assay kit (Catalog Number BC4760), β-galactosidase (β-GAL) Activity Assay Kit (Catalog Number BC2580), β-glucosidase (β-GC) Activity Assay Kit (Catalog Number BC2560), and 1,4-β-D-Glucan Cellobilhydrolase (C1) Activity Assay Kit (Catalog Number BC4300) (Beijing Solarbio Science & Technology Co., Ltd., Beijing, China), respectively, and quantified using an automatic microplate reader (Shanghai flash spectrum Biotechnology Co., Ltd., Shanghai, China).

### 4.3. Microscopic and Immunofluorescence Analyses

To visualize the hyphae in *U. maydis*-infected plant leaves, the focal tissue below the infection site was collected and fixed in FAA solution. The fungal and plant cell walls were co-stained using 4’,6-diamidino-2-phenylindole (DAPI; Sigma-Aldrich, St. Louis, MO, USA) and WGA-AF 488 (Thermo Fisher, Austin, TX, USA) as described by Gao [62] with some modifications including increased concentration and time. The images were obtained using a confocal microscope (ZEISS, LSM780) and analyzed using ZEN 3.4 software (Zeiss, Dublin, CA, USA).

The maize leaves infected with *U. maydis* were treated as described previously with increased dehydration time [63] for paraffin sectioning. Paraffin sections were prepared using a manual microtome (Leica RM2235, Wetzlar, Germany) and stained with 0.025% (*w*/*v*) toluidine blue. The progress and characteristics of *U. maydis* infection in the treatment and control groups of Ye478 and R445 were assessed using a Photometrics SenSys CCD camera and a stereomicroscope (ZEISS Axio Zoom.V16, Oberkochen, Germany).

Pectin, hemicelluloses, and cell wall glycoproteins were qualitatively analyzed via immunohistochemical staining. The monoclonal antibodies, namely, LM10 (for xylan), JIM5 (for low-methylated pectin), and LM20 (for high-methylated pectin) were used for immunolabeling. Cellulose was visualized via staining with 1% Calcofluor White (Fluorescent Brightener 28, Sigma, MO, USA) as previously described by Tang [9]. Immunofluorescence was observed under a Zeiss microscope (ZEISS, LSM780) with ZEN 3.4 software (Zeiss). NanoZoomer Digital Pathology RS (Pannoramic^®^ 250 Flash III, 3DHISTECH Ltd., Budapest, Hungary) was used to obtain a panoramic photo, and fluorescence signal intensity was regulated using Caseviewer (V2.3) software (DanJier Co., Ltd., Jinang, China). The photographed images were analyzed using Image J (National Institutes of Health, Kansas City, MO, USA) software, and the average fluorescence intensity was calculated as the average fluorescent point pixel value of each target site.

### 4.4. RNA Extraction and qRT-PCR

Total RNA was extracted from the leaves collected at 0, 2, 4, 8, and 12 dpi using an RNA extraction kit (DP432; TianGen Biotech, Beijing, China) as per the manufacturer’s protocol. The quantity and quality of RNA was confirmed using a NanoDrop 2000 (Thermofisher, MA, USA) according to A260/A280 and electropherograms (Appendix A; Appendix A). Complementary DNA (cDNA) was synthesized using a reverse transcription kit (Catalog Number RR047A, Takara Biotechnology, Dalian, China) as per the manufacturer’s instructions. Three differentially expressed genes (DEGs) were obtained from the transcriptome data, fungus-specific (ppi) primers were performed for the quantification of fungal biomass [2], and qRT-PCR with three technique and three biological replications were analyzed on a Bio-Rad CFX96 real-time PCR system (Bio-Rad, Richmond, CA, USA), and the reaction volume was 21.0 μL, 2 × SuperReal PreMix Plus 10 μL, foward primer 1.0 μL, reverse prime 1.0 μL, cDNA 2.0 μL, RNase Free ddH_2_O 7.0 μL. The 2^−ΔΔCt^ method was used to process the data. The method was modified by Zou [1]. MegaX software and Tbtools were used to analyze the sequence and function of the pectin methyl esterase-related genes of maize, Arabidopsis thaliana, and rice. And deep investigation was performed on the previous transcriptome data [1]. Primers were designed using NCBI, Primer 6, and Maize GDB web pages. 

The primers used are described in Appendix A. The inter-group differentiation between susceptible, partially susceptible, and resistant was performed using an LSD test.

## 5. Conclusions

In summary, after *U. maydis* infection, both susceptible and resistant lines exhibited an increased content of cellulose, hemicellulose, pectin, and lignin as revealed via qualitative and quantitative analyses. However, the activity of PME increased. As it was competitively symbiotic, *U. maydis* also produced CWDEs to degrade cellulose, hemicellulose, and pectin while the fungal biomass was significantly increased at 8 dpi and 12 dpi. The degree of pectin methyl esterification affected the rheological properties of the cell wall and affected the disease resistance of plants. The PME activity significantly increased in the susceptible lines, followed by partially susceptible and resistant lines. These results revealed that the resistance to *U. maydis* infection is correlated with the degree of pectin methyl esterification. This study enhanced the understanding of the interaction between the cell wall and pathogenic basidiomycetes during plant defense mechanisms.

## Figures and Tables

**Figure 1 ijms-24-14737-f001:**
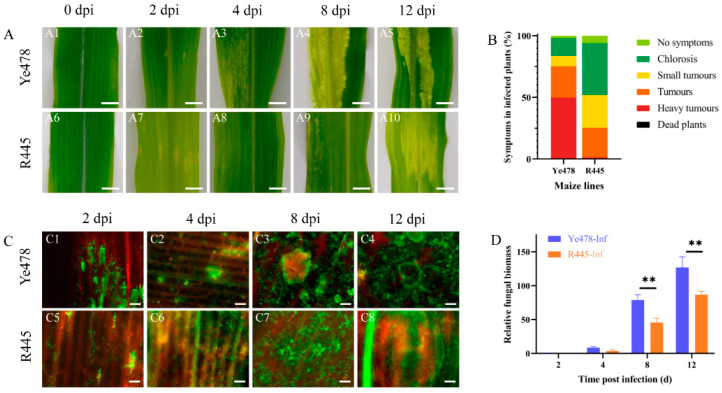
Phenotypic changes in maize after *U. maydis* infection. (**A**): Phenotypic changes in maize after *U. maydis* infection. (A1–A5 and A6–A10) Phenotype of Ye478 and R445, respectively, at 0, 2, 4, 8, and 12 days postinfection (dpi) with *U. maydis* SG200 strain; scale bar = 1 cm. (**B**): symptoms in Ye478 and R445 infected with *U. maydis*. (**C**): Hyphal development of *U. maydis* SG200 in maize leaves as detected with WGA AF488/propidium iodide co-staining. (C1–C4 and C5–C8) The hyphal growth dynamics in Ye478 and R445, respectively. The samples were collected at 2, 4, 8, and 12 dpi; scale bar = 100 μm. (**D**): Quantification of fungal biomass according to the amount of genomic DNA using fungus-specific (ppi) primers. qPCR was performed at 2, 4, 8, and 12 dpi using plant-specific (GAPDH) and fungus-specific (ppi) primers. Data were analyzed using Student’s *t* test. ** represents significant difference at *p* ≤ 0.01.

**Figure 2 ijms-24-14737-f002:**
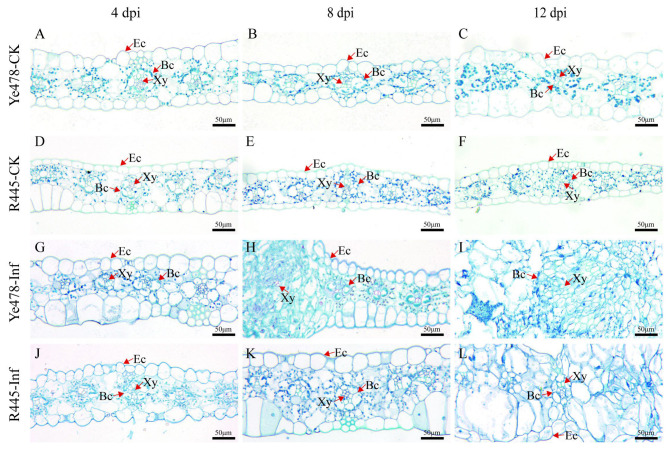
Histological sectioning of leaf tumors formed after *U. maydis* infection. (**A**–**C** and **D**–**F**) Leaf paraffin sections of Ye478-CK and R445-CK at 4, 8, and 12 dpi, respectively. (**G**–**I** and **J**–**L**) Leaf paraffin sections of Ye478-Inf and R445-Inf at 4, 8, and 12 dpi, respectively. Ec: Epidermal cell; Xy: Xylem; Bc: Bundle sheath cell. Scale bar = 50 μm.

**Figure 3 ijms-24-14737-f003:**
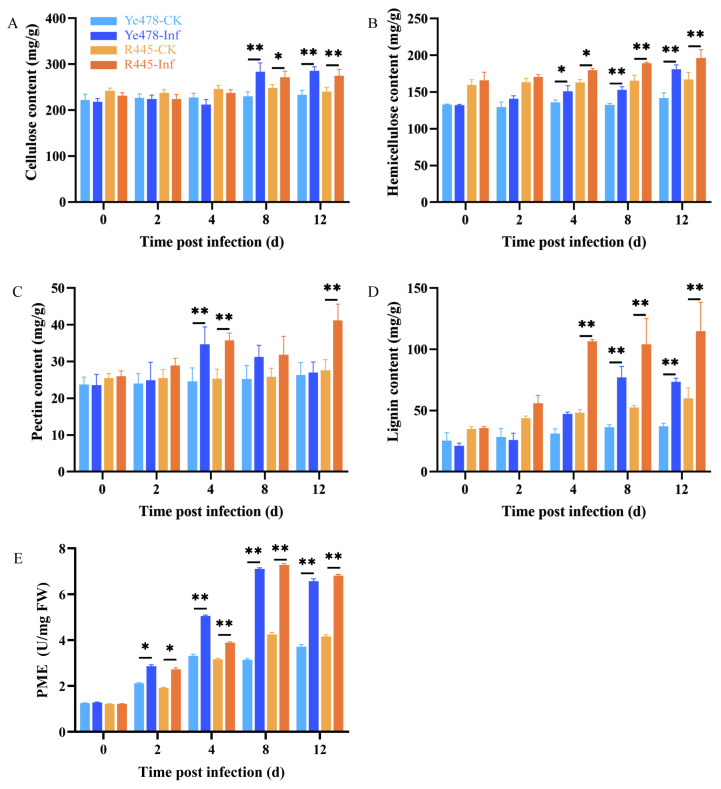
The changes of cell wall components in the leaves infected by *U. maydis*. (**A**) cellulose, (**B**) hemicellulose, (**C**) pectin, (**D**) lignin, and (**E**) pectin methylesterase (PME) activity in protein extracts from uninfected and infected leaves infected by *U. maydis*. Samples of Ye478 and R445 were collected at 0, 2, 4, 8, and 12 dpi. Ye478-CK/R445-CK: leaves infected with ddH_2_O; Ye478-Inf/R445-inf: leaves infected with *U. maydis*. Student’s *t* test, * and ** represent significant difference at *p* ≤ 0.05 and ≤0.01, respectively.

**Figure 4 ijms-24-14737-f004:**
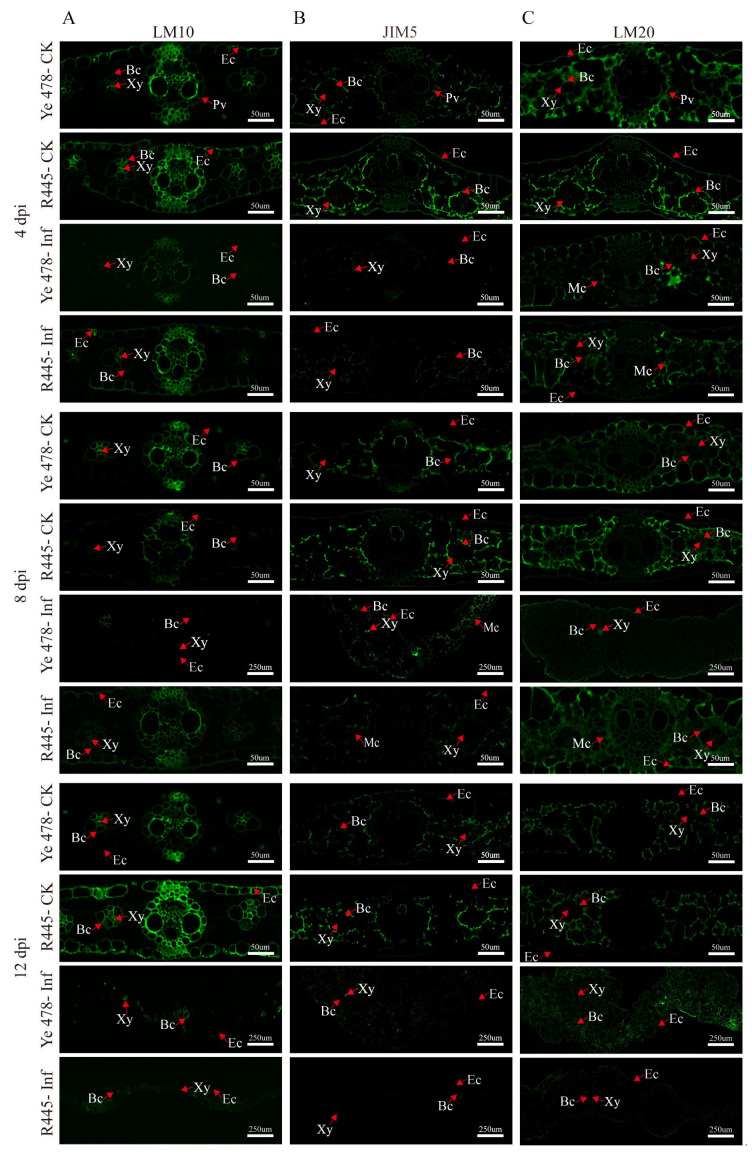
Immunolabeling of xylan and low/high-methylated pectin in maize leaves after *U. maydis* infection. Immune antibody markers including antibodies against plant xylan (LM10) (**A**), low-methylated pectin (JIM5) (**B**), and high-methylated pectin (LM20) (**C**) were used. Samples of Ye478 and R445 were collected at 4, 8, and 12 dpi. Ye478-CK: Ye478 leaves infected with ddH_2_O; Ye478-Inf: Ye478 leaves infected with *U. maydis* SG200; R445-CK: R445 leaves infected with ddH_2_O; R445-inf: R445 leaves infected with *U. maydis* SG200. Ec: Epidermal cell; Xy: Xylem; Bc: Bundle sheath cell; Mc: Mesophyll cell; Pv: Primary vein. Scale bar = 50 μm and 250 μm.

**Figure 5 ijms-24-14737-f005:**
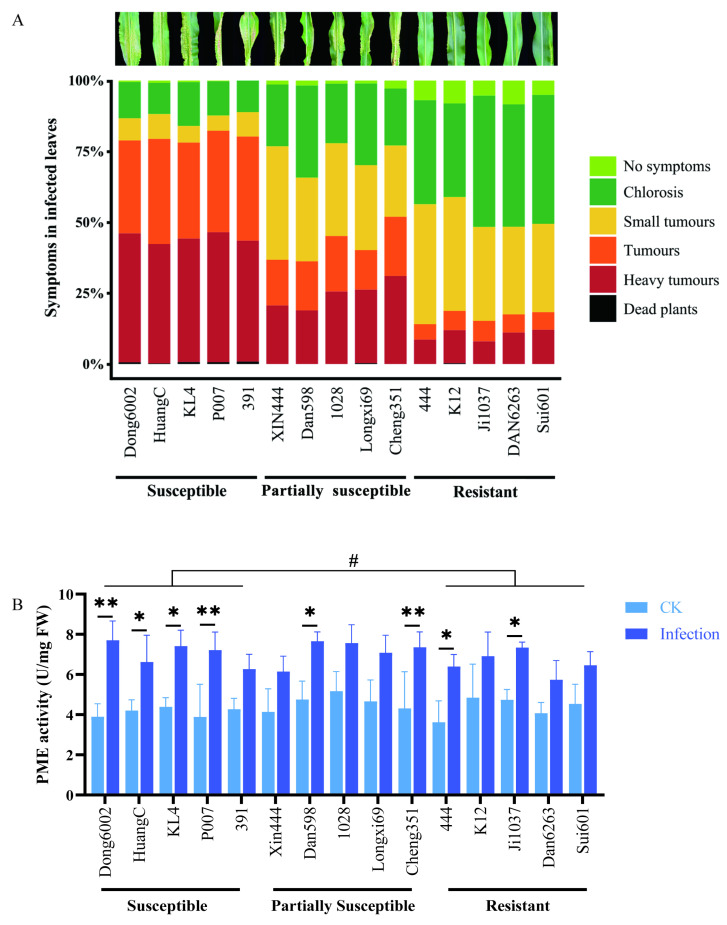
Development of *U. maydis* infection and PME activity in the 15 maize inbred lines with different resistant levels. (**A**) Disease symptom classification of 15 maize inbred lines with different resistant levels (susceptible, partially susceptible, and resistant) infected with *U. maydis* SG200. (**B**) Pectin methylesterase activity was quantified at 8 dpi. CK: inoculated ddH_2_O as control; Infection: leaves infected by *U. maydis*; Student’s *t* test, * and ** represent significant difference at *p* ≤ 0.05 and ≤0.01, respectively. LSD (least significant difference) test, ^#^ represent inter-group difference at *p* ≤ 0.05.

**Figure 6 ijms-24-14737-f006:**
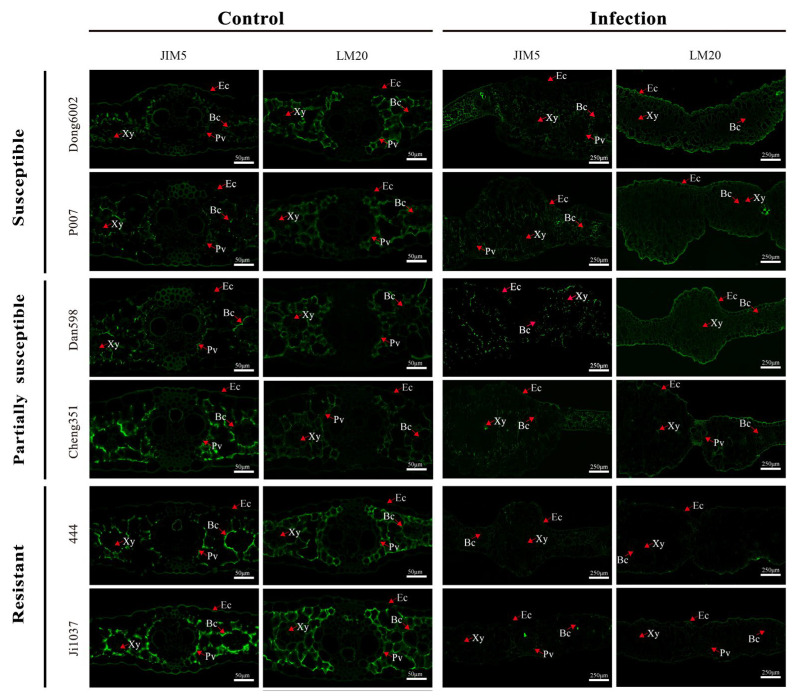
Immunolocalization of JIM5 and LM20 in the leaves of susceptible, partially susceptible, and resistant maize lines infected with *U. maydis* SG200. Control: inoculated ddH_2_O as control; Infection: leaves infected by *U. maydis*. Ec: Epidermal cell; Xy: Xylem; Bc: Bundle sheath cell; Pv: Primary vein. Scale bar = 50 μm (Control) and 250 μm (Infection).

**Figure 7 ijms-24-14737-f007:**
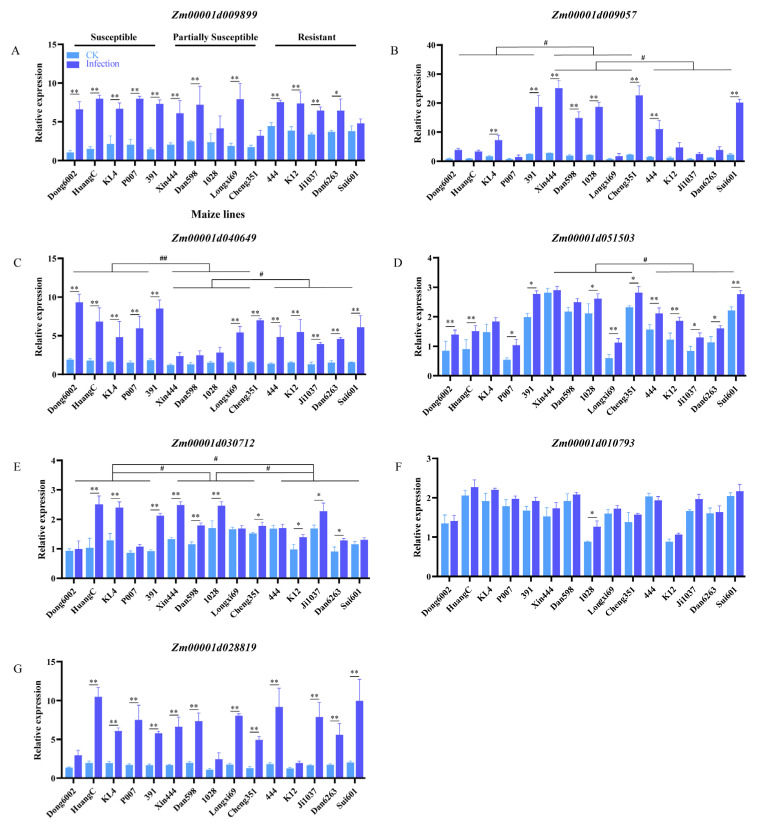
Expression of PME-synthesis-related genes during disease progression in 15 maize inbred lines with different resistant levels. Relative expression of seven PME-synthesis-related genes was assessed at 8 dpi using qRT-PCR. (**A**–**G**) PME-synthesis-related genes in maize. CK: inoculated ddH_2_O as control; Infection: leaves infected with *U. maydis* SG200; Student’s *t* test, * and ** represent significant difference at *p* ≤ 0.05 and ≤0.01, respectively. LSD (least significant difference) test, ^#^ and ^##^ represent inter-group difference at *p* ≤ 0.05 and ≤0.01.

## Data Availability

All data analyzed during this study are provided in this published article and Appendix A.

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
