# Peer review of "The Resistance of Maize to *Ustilago maydis* Infection Is Correlated with the Degree of Methyl Esterification of Pectin in the Cell Wall"

_ijms, 2023, doi:10.3390/ijms241914737_

Round 1

Reviewer 1 Report

The manuscript describes methyl esterification of pectin in the cell wall of Ustilago-infected maize. The dynamics of the leaf cell structure and the presence of high methoxyl pectin in the resistant variety have high significance in crop management strategies. I recommend the publication of this manuscript.

Author Response

Thank you very much for your comments. Wish you have a good day!

Reviewer 2 Report

The work investigates changes in cell wall composition in resistant and susceptible lines of maize at intervals during 12 days after infection by the pathogenic fungus Ustilago maydis. The investigation protocol included immunofluorescence detection of low- and high-methoxyl pectins, essaying of cell-wall-degrading enzymes of fungal origin and pectin methylesterases of the host, and host transcriptome analysis. From the results, the authors infer that pectin methoxylation plays a key role in maize resistance to U. maydis infection.

As expectable for a paper with over ten authors, the manuscript has a heterogeneous structure, with reasonably well written parts and clumsy sections.  More important, the results appear to be somewhat self-inconsistent and of difficult interpretation. Overall, the work supports the notion that pectin is involved in maize resistance to infection by U. maydis, yet it provides no clear evidence of a key role of pectin methoxylation. Admittedly, the authors are more cautious in the main text than foreseeable from the bold title they chose. Still, in my opinion the paper needs extensive revision starting from the title, which should be more neutral.

The most critical points are:

1.      Section 4 (Materials and Methods) should better explain which lines were investigated. My understanding is that most observations were performed on the Ye478 (susceptible) and R445 (resistant) line, whereas other lines with different degrees of resistance to fungal infection were analyzed only for PME activity and transcriptome.

2.      Lines 131-132: Section 4 does not include any information of the method applied for quantification of fungal biomass (mentioned in lines 132-132 and legend to Fig. 1.

3.      Subsection 2.2: Fig. 3E shows a steady decrease of PME activity in infected plants (both susceptible and resistant) from 0 to 12 dpi (and an incomprehensible increase in some controls). Inter alia, the main text refers to PME activity, whereas the figure refers to PME contents (expressed as ng/g FW). I wonder how the authors managed to quantify PME contents.

4.      Lines 168-169 It is not clear whether the changes observed are effects of infection or merely reflect leaf maturation. Why the contents of cellulose, hemicellulose, pectin, and lignin exhibited increase in the treatment and control groups of Ye478 and R445 as the dpi increased? Also doubtful are the results of immunofluorescence. For example, the immunolabelling of controls (CK) appears to change randomly (Fig. 2).

5.      Lines 224-226 From 4 to 12 dpi, immunolabelling with LM20 antibodies indicated that the signals were relatively stronger in Ye478-Inf and R445-Inf, particularly in the bundle sheath cells, epidermis, and mesophyll cells. According to Fig. 2, LM20 labelling markedly decreased from 4 to 12 dpi in both lines.

6.      Section 4 does not mention measurement of PME activity.

7.      Fig. 5B The asterisks only show the significance of the difference between controls and infected plants. More interesting would be the significance of the difference in PME activity between susceptible, partially susceptible, and resistant lines.

8.      Lines 262-263 The observation that immunolabelling with JIM5 antibody was weaker in all three groups than in the control puzzles me. Because the fungus elicits up-regulation of PMEs, I would anticipate that the JIM5 signal for low-methoxyl pectins were stronger in infected plants than in controls. This observation also applies to Fig. 2.

9.      Lines 265-267. Also puzzling is that immunolabelling with LM20 antibodies for high methoxyl pectins was stronger in susceptible and partially susceptible lines but weaker in resistant lines after the infection. If I understand the issue, up-regulation of PMEs should weakens LM20 labelling in susceptible lines more than in resistant lines. The figures provided do not help much as the fluorescence signal in infected samples is very weak throughout.

10.   Lines 287: No mention of Arabidopsis or rice in Section 4

11.   Subsection 2.6 and Fig 7. Although infection clearly upregulates the expression of host PME, there appears to be no significant difference between susceptible, partially susceptible and resistant lines. These results do support the notion that pectins are involved in maize interactions with Ustilago, but give no insight on the role of methoxylation. Also important, the results of transcriptome analysis (showing upregulation of PME genes) contrast with the analysis of PME activity (showing a decrease in infected plants, see point 3 above).

12.   Lines 567-569. The fact that after U. maydis infection both susceptible and resistant lines exhibited increased content of cellulose, hemicellulose, pectin, and lignin as revealed by qualitative and quantitative analyses, whereas at same time U. maydis produced CWDEs to degrade cellulose, hemicellulose, and pectin. This apparent paradox needs to be addressed.

Minor points:

Lines 82-85 need rewriting. In the present form the sentence is obscure and does not explain the nature of OGs, starting from the acronym.

Lines 151-152 “a blue-green lignified layer was observed 151 in the epidermis of R445-Inf but not in that of Ye478-Inf (Figure 2 G, J)” The magnification is too low. No difference is perceivable.

Line 547: No antibody against cell wall glycoprotein was employed in this study.

The file attached contains further suggestions in the form of notes. Parts that need special attention are marked.

Reviewer 3 Report

The study was focused on methyl esterification of pectin in the cell wall of maize confers resistance to Ustilago maydis. The Authors revealed that U. maydis infection altered pectin methyl esterification, and the degree of methyl esterification positively affected the resistance of maize to U. maydis. Furthermore, the relationship between pectin methyl esterification and host resistance was validated using 15 maize inbred lines with different resistant levels.

The paper is quite interesting, however, I recommend some significant improvements:

-        The Introduction is overloaded in the content, it should be presented in a more concise form.

-        Figure 5 – statistical analyses should be re-calculated. Factorial ANOVA with subsequent post-hoc test (e.g. Tukey’s test) is recommended because there are several variables tested.

-        Subchapter “4.4. RNA Extraction and qRT-PCR” in Materials and methods should be considerably revised. The house-keeping gene, Genbank accession numbers of tested genes, chemicals, real-time qRT-PCR assay, should be described in Materials and methods section.

-        I recommend including the electropherograms presenting the RNA bands in agarose gels in the manuscript or in the Supplementary file – it would provide information regarding quality of total RNA samples. The RNA concentration should be quantified at both 260 and 280 nm, and purity of RNA should be calculated based on the A260/A280 coefficient.

-        If Authors used SYBR Green fluorescent dye during RT-PCR gene expression studies, it is obligatory to perform Melting Curve Analysis, and results of this examination should be added in the manuscript or Supplementary file (e.g., JPG or TIFF file).

-        Statistical analyses paragraph should be included.

-        Extensive editing of English language is required.

Extensive editing of English language is required. 

Round 2

Reviewer 2 Report

The authors' conclusion, as expressed in the title of the paper, remains weakly supported by the data. Especially disturbing is that the authors refer to PME contents instead of activity, and assume that this explains the apparent incongruences in their data. Talking about enzymes, what realy matters is activity, not contents (which by the way is much more difficult to measure). By addressing many of my observations, however, the authors have significantly improved the paper.

English language use would benefit from further refinement
